# Optimizing Feeding Strategies for Growing Rabbits: Impact of Timing and Amount on Health and Circadian Rhythms

**DOI:** 10.3390/ani13172742

**Published:** 2023-08-28

**Authors:** Jie Huang, Qiangjun Wang, Kehao Zhang, Shuai He, Zhongying Liu, Mingyong Li, Man Liu, Yao Guo, Zhonghong Wu

**Affiliations:** 1State Key Laboratory of Animal Nutrition, College of Animal Science and Technology, China Agricultural University, Beijing 100193, China; huangjie3344@icloud.com (J.H.); wangqiangjun182@163.com (Q.W.); zhangkehao_cau@163.com (K.Z.); shuaihe2021cau@126.com (S.H.); lzy228@cau.edu.cn (Z.L.); 2National Rabbit Industry Technology System Qingdao Comprehensive Experimental Station, Qingdao 266431, China; lmy77@126.com (M.L.); liuman_18@163.com (M.L.)

**Keywords:** feeding pattern, growing rabbits, behavioral rhythm, body temperature rhythm, production performance, animal welfare

## Abstract

**Simple Summary:**

Mammals have circadian rhythms that regulate their behavior and physiology. Improper feeding can disrupt these rhythms and harm animal health. This study looked at the effects of feeding time and amount on growing rabbits in northern China during spring. Four groups of rabbits were given different feeding regimens, and the results showed that nighttime diet-unrestricted feeding had the best outcomes. This regimen improved the rabbits’ growth, circadian rhythm, and body temperature while reducing the risk of diarrhea and death. The group that had whole day diet-restricted feeding had higher daytime body temperatures but no significant difference in feed conversion rate. The study aimed to explore the impact of feeding timing (whole day vs. nighttime) and feeding amount (unrestricted vs. restricted) on the growth, behavior, circadian rhythm, and overall well-being of growing rabbits in northern China during spring. It is essential to consider feeding methods when raising animals to maintain their health and well-being.

**Abstract:**

Mammals exhibit circadian rhythms in their behavior and physiological activities to adapt to the diurnal changes of the environment. Improper feeding methods can disrupt the natural habits of animals and harm animal health. This study investigated the effects of feeding amount and feeding time on growing rabbits in northern China during spring. A total of 432 healthy 35-day-old weaned rabbits with similar body weight were randomly assigned to four groups: whole day diet-unrestricted feeding (WUF), whole day diet-restricted feeding (WRF), nighttime diet-unrestricted feeding (NUF), and nighttime diet-restricted feeding (NRF). The results showed that nighttime diet-unrestricted feeding improved performance, circadian rhythm of behavior, and body temperature, while reducing the risk of diarrhea and death. WRF group increased daytime body temperature but had no significant difference in feed conversion rate. The study suggests that nighttime diet-unrestricted feeding in spring can improve the growth and welfare of rabbits in northern China. Our study underscores the pivotal role of feeding timing in enhancing animal health. Future investigations should delve into the underlying mechanisms and expand the application of this strategy across seasons and regions to improve rabbit husbandry practices.

## 1. Introduction

New and efficient raising processes and precise managements are required urgently in the development of animal husbandry. In order to reduce labor costs and improve production efficiency, most of the current raising techniques and feeding patterns adopt intensive and automated feeding management. However, it is uncertain whether these techniques and feeding patterns align with the natural biological habits of livestock. The adaptation of animals to external photoperiods leads to a daily rhythm of behavior and physiological activities following a 24-h cycle. Disturbances to the feeding and activity cycle, stress, and sudden changes in environmental temperature can disrupt the biological rhythm of animals and interfere with metabolic homeostasis, and research has indicated that feeding them at night can result in decreased thermogenesis and increased fat deposition [1]. Feeding dairy cows at night alters their circadian rhythms, decreasing the circadian amplitude of body temperature, and disrupting the synthesis rhythm of milk fat, protein, and lactose, which shows an opposite trend to daytime feeding [2]. Our previous study also found that both pig raising with electronic feeding system and the artificial daytime feeding of rabbits are inconsistent with natural biological habits, which affected the production potential of livestock [3]. Studies on rodent animal models have shown that daytime eating can cause fat deposition, leading to obesity, diabetes, and intestinal inflammatory diseases [4]. However, aligning feeding habits with natural nocturnal patterns through nighttime feeding has been found to alleviate metabolic diseases caused by high-fat diets [5] and improve immunity [6]. These results suggest that inappropriate feeding patterns can disrupt endogenous biological rhythms, negatively affecting metabolic homeostasis. Feeding at improper times can cause mismatches between animal activity and feeding behavior rhythms, thereby disturbing the circadian rhythm of body temperature. Body temperature is a key zeitgeber for resetting the circadian clock, and it plays a role in regulating clock gene expression in peripheral tissues [7]. Our previous study found that the feeding time determined by raising technology and feeding patterns may not be consistent with the natural habits of domestic animals, which could affect the healthy and efficient raising of them. Our study in rabbits found that nighttime feeding in summer matched their eating and activity rhythm, enhanced circadian oscillation of body temperature, and promoted circadian expression of clock genes in liver and muscle, ultimately improving muscle growth [8]. Apart from feeding time, feeding amount also affects the biological rhythm of animals. It directly affects their sleep-wake cycle and indirectly regulates melatonin secretion and metabolic rate [9], ensuring their rhythm is synchronized with the environment. This emphasizes the significance of feeding animals at the appropriate time and with the right amount to promote their optimal growth and health.

Weaned rabbits are highly sensitive to environmental changes due to their underdeveloped immune function. The early spring season, with large diurnal air temperature difference, poses a significant challenge to young rabbits. The combination of low temperature environment and weaning stress increases the risk of diarrhea and death [10]. Previous studies have found that night-restricted feeding can promote the growth of rabbits, reducing mortality [8]. Our team also found that night-restricted feeding in summer significantly reduced the risk of death and diarrhea in rabbits [11]. These studies suggested that proper feeding pattern can effectively promote the healthy and efficient raising of rabbits without adding extra cost, and it also meets animal welfare requirements. However, it is unclear whether night-restricted feeding can improve rabbit performance in northern China during spring by enhancing the matching of feeding and activity rhythm with the circadian rhythm of body temperature, and this is a gap that needs to be addressed. The study aimed to explore the impact of feeding timing (whole day vs. nighttime) and feeding amount (unrestricted vs. restricted) on the growth, behavior, circadian rhythm, and overall well-being of growing rabbits in northern China during spring.

## 2. Materials and Methods

### 2.1. Experimental Animals and Feeding Management

The study was conducted in Qingdao, Shandong Province from 7 April to 25 May at an open rabbit farm facing east and west. A total of 432 healthy female rabbits, 35 days old with similar body weight (1.03 ± 0.07 kg), were selected for a two-factor randomized trial investigating feeding time and amount. The rabbits were randomly divided into four groups, with 108 rabbits in each group. The first two groups, whole day diet-unrestricted feeding (WUF) and whole day diet-restricted feeding (WRF), were fed at 6:30 a.m. and had access to food throughout the day. The other two groups, nighttime diet-unrestricted feeding (NUF) and nighttime diet-restricted feeding (NRF), were fed at 6:30 p.m. and could access food only during the 12-h nighttime period. The feeding schedule for each group is shown in Figure 1. All rabbits received the same diet with free access to water. The amount of remaining feed was measured at 6:30 am the next day. The feed amount was adjusted based on the growth curve and the health status of the rabbits during the experiment. The proportion of restricted feeding was consistent between the WRF and NRF groups each day [12]. All rabbits were raised in the same house with natural lighting, and the illumination time was determined by the sunrise and sunset times outside. Feed during the trial was supplied by Qingdao Kangda Feed Co., Ltd., Qingdao, China, with nutritional composition detailed in Table 1. The experimental protocol was approved by the Animal Ethics Committee of China Agriculture All University, Beijing, China.

### 2.2. Measurement of Production Performance Indicators

During the experiment, daily feed intake, feed residue, and incidence of mortality were recorded for all four groups, and the average daily feed intake was calculated. The rabbits in each group were weighed after a 12-h fast at 35, 42, 49, 56, 63, 70, 77, and 84 days of age, and the average daily weight gain and feed conversion rate were calculated for each stage. At the end of the experiment, the statistical analysis encompassed the average daily feed intake, average daily gain, and feed conversion rate of rabbits throughout the entire experimental period spanning from 36 to 84 days of age.

### 2.3. Experimental Environment, Behavior and Body Temperature Monitoring

Automatic dataloggers (179-TH, Apresys, Duluth, GA, USA) were utilized to record the dry bulb temperature and relative humidity both inside and outside the rabbit house. Meanwhile, an automatic water temperature recorder (179-T1L, Apresys, Duluth, GA, USA) was used to detect the drinking water temperature at 10-min intervals throughout the entire trial period. Additionally, the light intensity was measured hourly using an illuminometer (TES1332A, Taiwan, China). Prior to the commencement of the experiment, all instruments were tested for their accuracy and consistency. During the experiment, the deaths of rabbits were recorded every day, and the odds ratio of mortality was calculated. Rectal temperatures of the rabbits were measured continuously every two weeks at 1-h intervals, with 3 rabbits from each group.

Rabbit behavior was monitored using an infrared camera (ds-2PT7D20IW-DE, HIKVISION, Hangzhou, China) during the trial period. From 64 to 66 days of age, three rabbits from each group were selected for behavioral observation using analysis software (The Observer XT, Noldus, Wageningen, The Netherlands) to record the total time, frequency, and duration of activities such as eating, drinking, walking, and lying. The behaviour of rabbits is described as follows [13,14,15]. Regarding eating, the rabbit approaches the feed box, including the time with the head in the feeder, chewing, and swallowing. Regarding drinking, the rabbits touch the water spout with their mouths and keep their heads still. Regarding walking, the rabbits move from one place to another. Regarding grooming, licking, scratching, or nibbling the body or fur, including self-grooming and grooming of other rabbits occurs. Regarding resting, sitting, lying down, and stretching the body occurs.

### 2.4. Statistical Analysis

All environmental data were presented as mean ± SD, while performance, behavior, and rectal temperature data were presented as mean ± SEM. The odds ratio of diarrhea and death was analyzed using 95% confidence intervals. The non-parametric Jonckheere-Terpstra-Kendall cycle (JTK_Cycle) test in R software (https://r-project.org/, accessed on 1 July 2023) was used to evaluate the significance, phase, and amplitude of 24-h rhythms in behavior and abdominal temperature of rabbits. SPSS20.0 software was used to analyze data on performance, behavior, and body temperature by two-way ANOVA and Duncan multiple comparison analysis. Statistical significance was set at *p* < 0.05. GraphPad Prism (version 7.00, GraphPad software, San Diego, CA, USA) was used for figure analysis.

## 3. Results

### 3.1. Impact of Feeding Amount and Feeding Time on Rabbit Performance

The rabbits had access to drinking water with a temperature of 17.31 ± 2.93 °C during the experiment. The average indoor and outdoor temperatures were 17.55 ± 2.99 °C and 18.65 ± 3.50 °C, respectively. The indoor and outdoor relative humidity were 67.16 ± 3.50% and 60.66 ± 17.17% (Figure 2).

Comparing the effects of different feeding methods on rabbit production performance, two-way ANOVA showed significant effects of feeding time on daily weight gain, daily feed intake, and the feed conversion ratio of rabbits, except for the period between 64–70 days of age. The interaction between feeding amount and feeding time significantly affected the daily feed intake of rabbits, except for the period between 71–77 days of age (Table 2. *p* < 0.05). The body weight of rabbits in the WRF and WUF groups at 84 days of age was significantly higher than that in the NRF and NUF groups (*p* < 0.05). Additionally, the feed conversion ratio of the WUF group was significantly higher than that of the other groups (*p* < 0.05). The daily weight gain and daily feed intake of rabbits in the WRF and WUF groups were significantly higher than those in the NRF and NUF groups at 36–42 days and 71–84 days of age (*p* < 0.05). However, at 57–63 days of age, the daily weight gain of meat rabbits in the WRF group was significantly lower than that in the other three groups, while the feed conversion rate was significantly higher than that in the other three groups (Table 2). This may have been due to the sudden change in ambient temperature at this stage, which caused a decrease in daily gain.

### 3.2. Impact of Feeding Amount and Feeding Time on the Health of Rabbits

As shown in Figure 3, there was no significant difference in the risk of death between WRF and NRF groups under restricted feeding conditions. However, under unrestricted feeding conditions, the cumulative risk of death was more than twice that in the NUF group––between 63 to 84 days of age. Throughout the experiment spanning from 36 to 84 days of age, the mortality rates of 108 rabbits in each group were as follows: NRF, WRF, NUF, and WUF groups exhibited rates of 4.63%, 4.63%, 0.96%, and 2.78% respectively. These findings suggest that nighttime feeding aligns with the nocturnal feeding habits of rabbits under unrestricted feeding conditions, which is more favorable for the healthy raising of rabbits.

### 3.3. Impact of Feeding Amount and Feeding Time on Behavioral Rhythm of Rabbits

According to JTK analysis of circadian rhythms of feeding, walking, drinking, grooming, and resting behaviors of rabbits (Figure 4), eating and walking behaviors had significant circadian rhythms in all four groups (ADJ.*p* < 0.05). Drinking behavior had a significant circadian rhythm only in WRF group, and the peak value appeared at night (ADJ.*p* < 0.05). Through analysis of the peak time of each behavior in the four groups, it was revealed that the peak eating behavior in the NRF and NUF groups occurred in the evening, whereas in the WRF and WUF groups, it occurred in the morning. Additionally, the peak walking behavior in all four groups occurred during the nighttime, while the peak resting behavior occurred during the daytime, which is in line with the behavior of nocturnal rabbits.

Based on the analysis of the duration of various behaviors in rabbits (Table 3), diurnal differences in eating, drinking, walking, grooming, and lying behaviors were observed in the NRF group under restricted feeding conditions, whereas only grooming and lying behaviors exhibited diurnal differences in the WRF group. Compared to the WRF group, the NRF group exhibited mainly nighttime eating, drinking, and walking behaviors, with a decrease in total eating time and an increase in walking time (*p* < 0.05). Additionally, grooming and lying behaviors significantly increased in the NRF group during daytime (*p* < 0.05).

Under unrestricted feeding conditions, NUF and WUF groups exhibited significant diurnal differences in eating, walking, grooming, and lying behaviors, but no significant difference in the total duration of these behaviors was observed. The NUF group exhibited a significant reduction in eating time during the daytime, with a significant increase at nighttime, and a significant increase in grooming time during the daytime compared to the WUF group. Analysis of commonalities among the four groups revealed significantly higher grooming behavior during nighttime than daytime (*p* < 0.05), and significantly higher lying behavior during daytime than nighttime (Table 3. *p* < 0.05), further supporting the nocturnal characteristics of domestic rabbits.

### 3.4. Impact of Feeding Amount and Feeding Time on Rectal Temperature of Rabbits

According to Figure 5, the rectal temperature of rabbits in the WNF, NUF and NRF groups exhibited a diurnal fluctuation pattern with a decrease during the daytime and an increase during the nighttime. However, in contrast, the rectal temperature of rabbits in the WRF group peaked at dusk, and showed diurnal fluctuations with high values during the daytime and low values at nighttime. Furthermore, we compared the effects of feeding time on body temperature amplitude under both restricted and unrestricted feeding amount conditions. The results showed that the daily temperature amplitude of the NUF group was higher than that of the WUF group, and that of the NRF group was higher than that of the WRF group. These findings suggest that nighttime feeding aligns with the circadian rhythm, enhancing diurnal body temperature fluctuation and amplitude. Feeding amount has a moderate effect on diurnal temperature fluctuation but has little impact on amplitude.

## 4. Discussion

The dynamic balance between heat dissipation and thermogenesis is the key to maintaining a relatively constant body temperature in endotherm. In fact, the body temperature of endotherms exhibits circadian oscillations within the physiological range [7]. Studies on nocturnal animals, such as mice, have found that body temperature shows the circadian rhythm increasing during nighttime and decreasing during daytime [16,17], and the circadian oscillation of body temperature is mainly determined by the sympathetic activity, eating-induced heat increment [18], activity-induced muscle thermogenesis, and ambient temperature [19]. Previous studies have found that the sympathetic activity regulated by SCN can rhythmically regulate thermogenesis of both brown adipose tissue and muscle activity, forming the diurnal oscillation of body temperature [20,21]. Decreased sympathetic activity during early sleep lowered body temperature, while increased sympathetic activity during wakefulness raised body temperature [21]. The night feeding matched with the natural habit of rabbits and can promote thermogenesis by up-regulating the expression of brown adipose uncoupling protein (UCP1). On the other hand, activity-induced skeletal muscle contraction leads to a significant release of Ca^2+^ from the sarcoplasmic reticulum, which activates AMPK [22]. Activation of AMPK can cause the deacetylation of PGC-1α, regulating the expression of the clock gene BMAL1 [23], and accelerating muscle triglyceride breakdown, providing energy for thermogenesis [24]. In this study, NUF and NRF promoted eating and walking behavior that mainly occurred at nighttime, which may promote the superposition of skeletal muscle heat production and heat increment at night, thereby enhancing the rhythmic fluctuation of rabbit body temperature between lower at daytime and higher at nighttime. These results suggest that feeding patterns matching the natural eating habits of rabbits can promote the circadian fluctuations of body temperature.

As an important zeitgeber, temperature fluctuations can reset the expression of clock genes in peripheral tissues [25,26]. Our previous studies have shown that night feeding enhances the rhythmic fluctuation of body temperature, which in turn regulates the expression of muscle clock genes through the HSF1 nuclear receptor pathway, thereby promoting muscle protein synthesis by increasing the phosphorylation of mTOR and S6K [26,27]. Consistent with the results of previous studies, NUF and NRF promote rabbits to feed at night and increase nocturnal walking behavior, which may enhance the body temperature of rabbits at night through strengthening the skeletal muscle heat production and heat increment so as to resist the adverse effects of a low temperature environment by increasing body temperature at night. In addition, the grooming behavior of domestic rabbits can serve as an important indicator to assess their comfort level. When rabbits feel comfortable and relaxed, they typically engage in grooming behavior, which helps them to stay dry and clean. In this study, rabbits in the NUF group exhibited the longest grooming duration, indicating that NUF has improved the comfort level of domestic rabbits.

Circadian oscillation of body temperature not only plays a role in regulating the expression of muscle clock genes but also changes the structure and composition of intestinal microflora [28]. The diurnal variation of intestinal microbial flora can regulate the rhythmic expression of tight junction proteins CLAUDIN-1 and OCCLUDIN mediated by intestinal clock genes through metabolites, such as short-chain fatty acids, serotonin, and bile acids, thus affecting the intestinal barrier function [29,30]. Therefore, the circadian oscillation of body temperature can improve the intestinal barrier function and reduce diarrhea and mortality in livestock [31,32,33]. In this study, the cumulative risk of death and diarrhea was analyzed in four groups of rabbits. It was revealed that NUF enhanced the circadian oscillation of body temperature and tended to reduce the risk of death. In addition, maintaining the intestinal health of the host is crucial for promoting the digestion and absorption of nutrients. Our previous research found that drinking water temperature affects intestinal flora composition and the intestinal health status of producing rabbits [11,34]. In this study, the average indoor temperature of the rabbit house was lower than the appropriate temperature required by rabbits (20–25 °C), and the temperature and water temperature in the house at night were lower than in the daytime. It was speculated that NUF promoted rabbits to consume more energy in order to resist the adverse effects of lower temperature and water temperature by strengthening muscle activity to produce heat. Due to the restricted feed amount, NUF group rabbits could not eat enough feed to supply energy at night, thus reducing the performance of rabbits. During the 60–62 days of age, the daily gain of the WUF group decreased significantly due to the sudden drop indoor temperature and drinking water temperature, which was consistent with the previous report that cold air temperature in rabbit farms caused a decrease in feed intake after the sudden drop of indoor temperature. Growing rabbits need higher ambient temperatures to maintain body temperature balance. However, the temperature difference between day and night is large in spring in northern China, so the restricted feeding, no matter whether it occurs in the daytime or nighttime, is not conducive to maintaining the circadian rhythm of body temperature, and this is not conducive to the healthy and efficient raising of rabbits.

## 5. Conclusions

The nocturnal feeding method employed in open rabbit houses during northern China’s spring aligns with the natural behavior of nocturnal rabbits. This reinforces their circadian rhythm, enhancing diurnal body temperature fluctuations and feed conversion efficiency, thereby promoting effective and healthy rabbit rearing. This study addresses the research gap in understanding the impact of feeding timing and quantity on rabbit growth, behavior, circadian rhythm, and overall well-being in northern China during spring. The authors’ findings emphasize the pivotal role of feeding timing in optimizing animal health. Further exploration of the underlying mechanisms and broader implementation of this feeding strategy across diverse seasons and regions are recommended to enhance rabbit husbandry practices.

## Figures and Tables

**Figure 1 animals-13-02742-f001:**
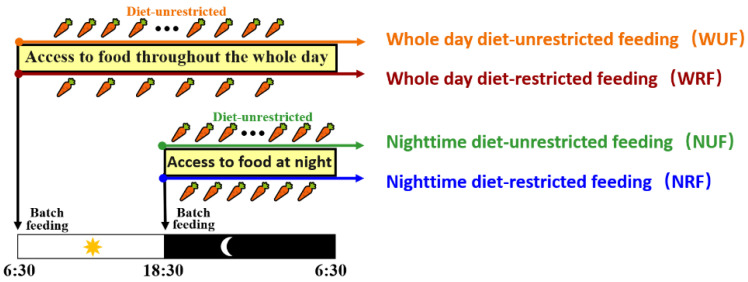
Schematic representation of the four feeding regimens used in this study (*n* = 432). NUF and NRF rabbits were allowed access to food from 6:30 p.m. to 6:30 a.m. the next day. WUF and WRF rabbits were allowed access to food during the whole day.

**Figure 2 animals-13-02742-f002:**
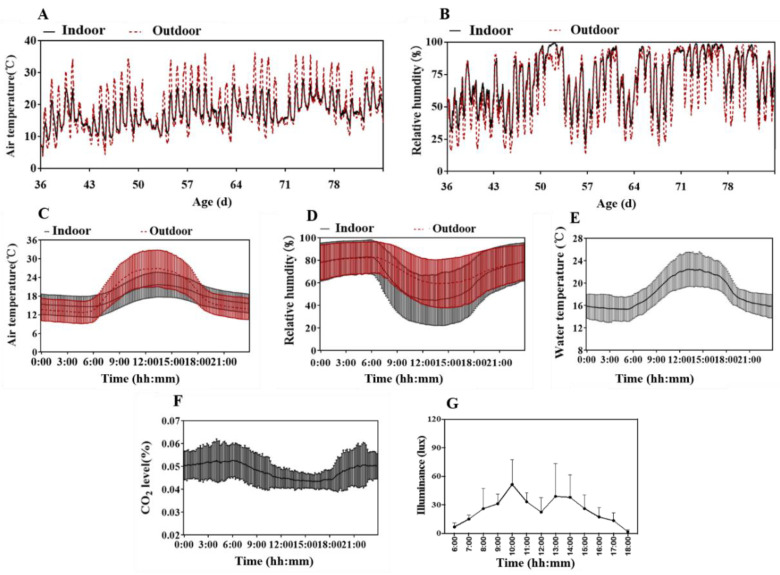
Thermal environment parameters during experiment for the rabbit house. (**A**) The mean indoor and outdoor air temperature. (**B**) The mean indoor and outdoor relative humidity. (**C**,**D**) The air temperature and relative humidity change indoor and outdoor over 24 h. (**E**) The drinking water temperature of growing rabbits. (**F**) The indoor CO_2_ concentration. (**G**) The indoor light intensity of each layer. Data are presented as mean ± SD.

**Figure 3 animals-13-02742-f003:**
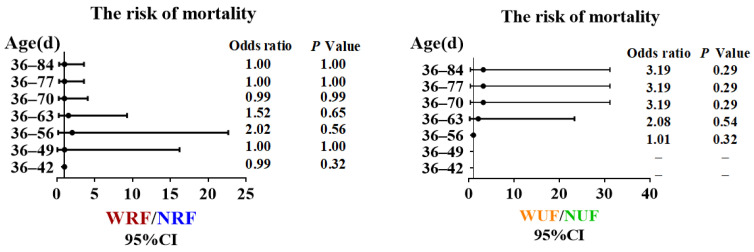
Feeding amount and feeding time influences mortality risk. Odds ratios were calculated with 95% confidence intervals.

**Figure 4 animals-13-02742-f004:**
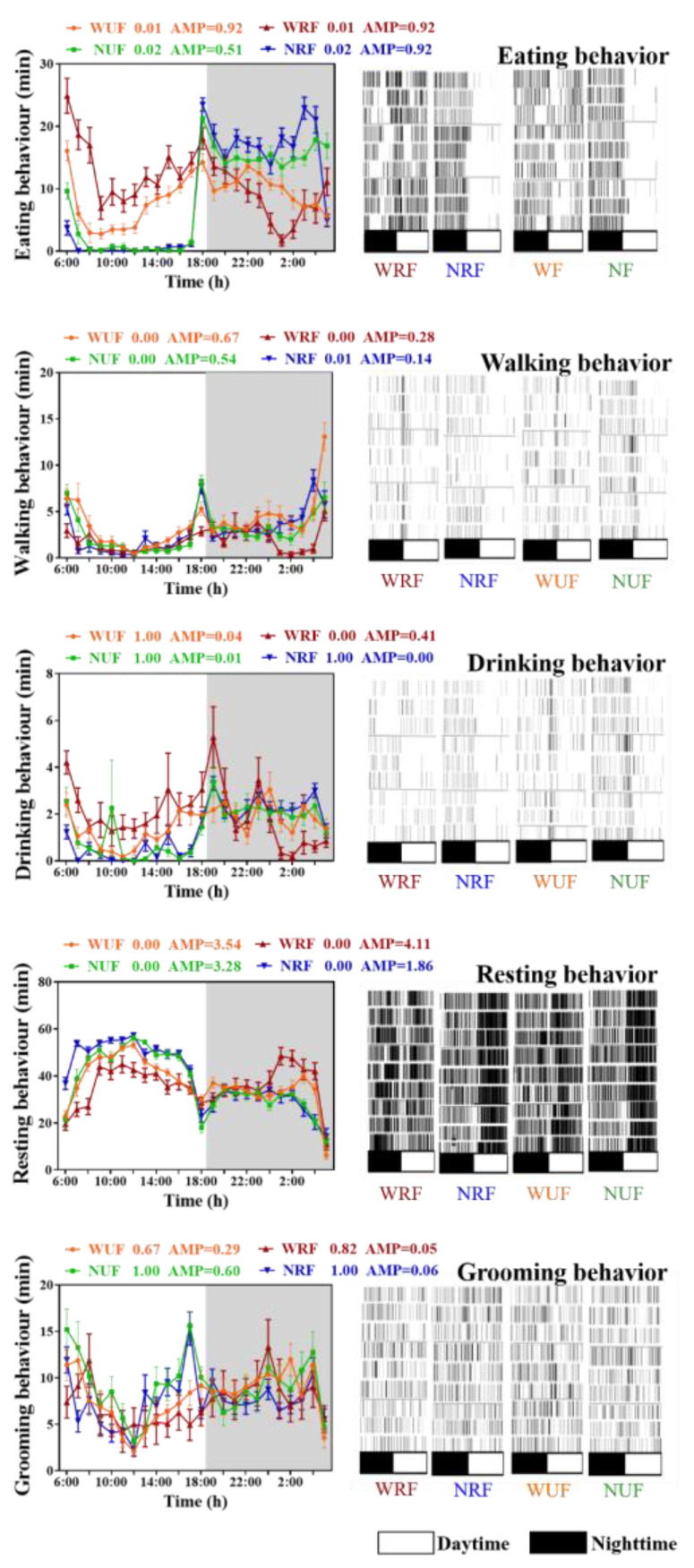
Feeding amount and feeding time influences behaviors rhythm. The diurnal rhythms based on JTK analysis, ADJ.*p* for adjusted minimal *p*-values, ADJ.*p* < 0.05 indicates a significant effect on circadian rhythm, and AMP represents amplitude. White in the graph represent daytime, and gray represent nighttime. *n* = 3 over 3 days. Data are presented as mean ± SEM.

**Figure 5 animals-13-02742-f005:**
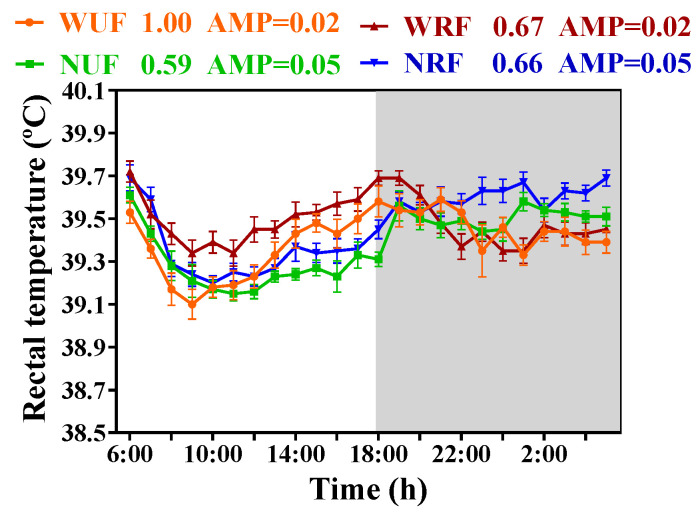
Feeding amount and feeding time influences the rhythm of rectal temperature. The diurnal rhythms based on JTK analysis: ADJ.*p* for adjusted minimal *p*-values, ADJ.*p* < 0.05 indicates a significant effect on circadian rhythm, and AMP represents amplitude. White in the graph represents daytime, and gray represents nighttime. *n* = 3 over 3 days. Data are presented as mean ± SEM.

**Table 1 animals-13-02742-t001:** Base feed raw material and nutrient content.

Raw Materials	Content	Nutritional Ingredients	Content
Alfalfa meal	31.80	Dry matter (g/kg feed)	868.00
Corn	27.60	Crude protein (g/kg DM)	202.00
Soybean meal	17.70	Neutral detergent fibre (g/kg DM)	318.00
Wheat bran	20.00	Ether extract (g/kg DM)	30.00
Premix	0.15	Digestible energy (MJ/kg DM)	11.89
Salt	0.40	Ca (g/kg DM)	10.40
Limestone	0.39	Total phosphorus (g/kg DM)	7.40
DL-Methionine	0.41	Lysine (g/kg DM)	9.80
Lysine	0.05	Methionine + cysteine (g/kg DM)	9.30
Calcium hydrophosphate	1.30	Threonine (g/kg DM)	9.20
Threonine	0.2		
Total	100.00		

Note: Premix provided per kg of diet: 12,000 IU of vitamin A; 2500 IU of vitamin D3; 40 mg of vitamin E; 2.0 mg of vitamin K; 2.0 mg of vitamin B1; 4 mg of vitamin B2; 2.0 mg of vitamin B6; 0.01 mg of vitamin B12; 0.06 mg of biotin; 50 mg of niacin; 0.3 mg of folic acid; 10 mg of D-pantothenic acid; 1000 mg of choline; 40 mg of Zn; 10 mg of Cu; 30 mg of Mn; 50 mg of Fe; 0.5 mg of I; 0.2 mg of Se; 0.5 mg of Co.

**Table 2 animals-13-02742-t002:** Effects of feeding amount and time on rabbit growth.

Performance Indicators	Age	WRF	NRF	WUF	NUF	*p* Value		
						Feed Amount	Feed Time	Feed Amount × Feed Time
Body weight (kg)	35	1.03 ± 0.00 ^a^	1.02 ± 0.00 ^a^	1.03 ± 0.00 ^a^	1.02 ± 0.00 ^a^	0.837	0.057	0.898
	42	1.30 ± 0.01 ^b^	1.27 ± 0.01 ^c^	1.33 ± 0.01 ^a^	1.25 ± 0.01 ^d^	0.581	<0.001	0.001
	49	1.64 ± 0.01 ^a^	1.54 ± 0.01 ^c^	1.65 ± 0.01 ^a^	1.59 ± 0.01 ^b^	0.001	<0.001	0.060
	56	1.95 ± 0.01 ^a^	1.84 ± 0.01 ^c^	1.96 ± 0.01 ^a^	1.90 ± 0.01 ^b^	0.014	<0.001	0.024
	63	2.14 ± 0.01 ^b^	2.14 ± 0.01 ^b^	2.21 ± 0.01 ^a^	2.18 ± 0.01 ^a^	<0.001	0.223	0.197
	70	2.43 ± 0.01 ^bc^	2.42 ± 0.01 ^c^	2.49 ± 0.01 ^a^	2.47 ± 0.01 ^ab^	<0.001	0.240	0.582
	77	2.75 ± 0.01 ^ab^	2.71 ± 0.01 ^b^	2.77 ± 0.02 ^a^	2.74 ± 0.02 ^ab^	0.077	0.016	0.827
	84	3.10 ± 0.02 ^a^	3.03 ± 0.02 ^b^	3.14 ± 0.02 ^a^	3.05 ± 0.02 ^b^	0.219	<0.001	0.567
Daily gain weight (g)	36–42	38.84 ± 0.74 ^b^	35.18 ± 0.87 ^c^	42.39 ± 0.8 ^a^	32.73 ± 0.85 ^d^	0.415	<0.001	<0.001
	43–49	48.54 ± 0.74 ^a^	37.79 ± 0.86 ^b^	46.44 ± 0.77 ^a^	47.47 ± 0.75 ^a^	<0.001	<0.001	<0.001
	50–56	44.91 ± 0.81 ^a^	45.37 ± 0.71 ^a^	42.26 ± 0.98 ^b^	46.01 ± 0.77 ^a^	0.207	0.014	0.043
	57–63	26.32 ± 0.96 ^c^	42.81 ± 0.84 ^a^	36.35 ± 0.86 ^b^	41.24 ± 0.82 ^a^	<0.001	<0.001	<0.001
	64–70	40.62 ± 0.88 ^ab^	41.63 ± 0.69 ^ab^	39.41 ± 0.73 ^b^	42.59 ± 0.76 ^a^	0.969	0.005	0.167
	71–77	45.18 ± 1.03 ^a^	40.82 ± 0.77 ^bc^	42.67 ± 0.80 ^b^	38.90 ± 0.90 ^c^	0.018	<0.001	0.504
	78–84	49.37 ± 0.88 ^b^	46.45 ± 0.93 ^c^	52.45 ± 1.16 ^a^	43.92 ± 1.12 ^c^	0.652	<0.001	0.018
	36–84	41.66 ± 0.47 ^b^	41.23 ± 0.29 ^b^	42.96 ± 0.42 ^a^	41.26 ± 0.40 ^b^	0.088	0.010	0.149
Daily feed intake (g)	36–42	97.29 ± 0.16 ^b^	90.79 ± 0.51 ^c^	113.69 ± 0.79 ^a^	87.63 ± 0.75 ^d^	<0.001	<0.001	<0.001
	43–49	114.02 ± 0.38 ^c^	114.43 ± 0.23 ^c^	136.9 ± 1.04 ^a^	127.42 ± 1.02 ^b^	<0.001	<0.001	<0.001
	50–56	145.63 ± 0.34 ^b^	137.66 ± 0.79 ^c^	151.81 ± 1.17 ^a^	150.46 ± 0.93 ^a^	< 0.001	<0.001	<0.001
	57–63	171.30 ± 0.75 ^b^	173.248 ± 0.56 ^b^	176.74 ± 1.53 ^a^	173.50 ± 1.19 ^b^	0.785	0.002	<0.001
	64–70	195.60 ± 0.45 ^a^	191.31 ± 0.78 ^b^	191.82 ± 1.12 ^b^	194.74 ± 0.99 ^a^	0.845	0.437	<0.001
	71–77	205.62 ± 0.61 ^a^	198.92 ± 1.14 ^b^	200.22 ± 1.32 ^b^	193.63 ± 1.29 ^c^	<0.001	<0.001	0.962
	78–84	214.57 ± 0.50 ^b^	207.28 ± 1.05 ^c^	221.13 ± 1.61 ^a^	200.78 ± 1.35 ^d^	0.983	<0.001	<0.001
	36–84	162.24 ± 0.29 ^b^	157.57 ± 0.49 ^d^	168.76 ± 0.69 ^a^	160.07 ± 0.78 ^c^	<0.001	<0.001	0.001
Feed conversion ratio	36–42	2.61 ± 0.06 ^a^	2.81 ± 0.09 ^a^	2.67 ± 0.05 ^a^	2.77 ± 0.07 ^a^	0.883	0.025	0.462
	43–49	2.44 ± 0.05 ^d^	3.08 ± 0.07 ^a^	2.93 ± 0.05 ^b^	2.62 ± 0.04 ^c^	0.785	0.002	<0.001
	50–56	3.33 ± 0.07 ^b^	3.09 ± 0.08 ^c^	3.80 ± 0.09 ^a^	3.34 ± 0.07 ^b^	<0.001	<0.001	0.162
	57–63	6.94 ± 0.26 ^a^	4.24 ± 0.11 ^c^	4.95 ± 0.11 ^b^	4.26 ± 0.09 ^c^	<0.001	<0.001	<0.001
	64–70	4.65 ± 0.09 ^b^	4.68 ± 0.08 ^b^	5.05 ± 0.10 ^a^	4.78 ± 0.10 ^b^	0.007	0.193	0.101
	71–77	4.63 ± 0.10 ^b^	4.99 ± 0.10 ^a^	4.87 ± 0.10 ^ab^	5.04 ± 0.11 ^a^	0.160	0.011	0.353
	78–84	4.30 ± 0.09 ^b^	4.47 ± 0.10 ^ab^	4.28 ± 0.11 ^b^	4.61 ± 0.11 ^a^	0.506	0.004	0.354
	36–84	3.87 ± 0.05 ^b^	3.83 ± 0.02 ^b^	3.97 ± 0.04 ^a^	3.86 ± 0.03 ^b^	0.041	0.026	0.218

Note: Different lowercase letters mean significant difference in different treatments at the same age (*p* < 0.05). Data are presented as mean ± SD, *n* = 108.

**Table 3 animals-13-02742-t003:** Effects of feeding amount and feeding time on behaviors rhythm.

Behaviors	Time Period (min/24 h)	WRF	NRF	WUF	NUF	*p* Value		
						Feed Amount	Feed Time	Feed Amount × Feed Time
Eating	Whole day	248.06 ± 6.97 ^a^	184.47 ± 6.61 ^b^	192.04 ± 6.10 ^b^	191.27 ± 5.80 ^b^	0.064	0.017	0.020
	Daytime	134.88 ± 2.82 ^aA^	9.21 ± 0.80 ^cB^	76.86 ± 1.47 ^bB^	20.96 ± 1.47 ^cB^	<0.001	<0.001	<0.001
	Nighttime	119.26 ± 2.51 ^bA^	175.26 ± 3.94 ^aA^	115.19 ± 3.19 ^bA^	170.31 ± 3.25 ^aA^	0.962	<0.001	0.629
Drinking	Whole day	38.04 ± 2.16 ^a^	27.84 ± 1.09 ^a^	32.56 ± 2.27 ^a^	35.73 ± 3.37 ^a^	0.801	0.463	0.167
	Daytime	16.81 ± 0.66 ^aA^	4.52 ± 0.17 ^bB^	12.44 ± 0.74 ^abA^	11.90 ± 1.56 ^abA^	0.570	0.020	0.032
	Nighttime	25.79 ± 1.33 ^aA^	23.32 ± 0.68 ^aA^	20.11 ± 1.08 ^aA^	23.42 ± 1.37 ^aA^	0.380	0.897	0.397
Walking	Whole day	54.64 ± 2.42 ^c^	85.66 ± 7.14 ^b^	131.32 ± 7.8 ^a^	121.88 ± 5.62 ^a^	<0.001	0.631	0.002
	Daytime	21.76 ± 1.75 ^bcA^	28.79 ± 0.48 ^cB^	46.36 ± 2.25 ^aB^	47.27 ± 2.09 ^bB^	<0.001	0.549	0.003
	Nighttime	33.90 ± 3.47 ^bA^	45.85 ± 5.15 ^bA^	84.96 ± 3.49 ^aA^	75.55 ± 3.11 ^abA^	0.221	0.163	0.027
Grooming	Whole day	158.33 ± 24.10 ^a^	175.54 ± 5.11 ^a^	173.09 ± 3.04 ^a^	216.12 ± 13.74 ^a^	0.337	0.296	0.652
	Daytime	24.12 ± 3.29 ^cB^	76.48 ± 1.36 ^bB^	71.80 ± 2.25 ^bB^	99.44 ± 4.07 ^aB^	0.002	0.001	0.760
	Nighttime	134.21 ± 16.57 ^aA^	99.06 ± 1.53 ^aA^	101.29 ± 2.49 ^aA^	116.28 ± 6.23 ^aA^	0.505	0.407	0.459
Lying	Whole day	1005.38 ± 9.02 ^a^	978.11 ± 13.39 ^ab^	902.13 ± 10.24 ^b^	904.46 ± 16.54 ^b^	0.001	0.625	0.562
	Daytime	520.04 ± 2.97 ^bA^	607.59 ± 2.66 ^aA^	512.23 ± 2.02 ^bA^	553.92 ± 6.71 ^bA^	0.004	0.001	0.020
	Nighttime	485.34 ± 9.23 ^aB^	370.52 ± 7.03 ^abB^	389.90 ± 6.00 ^abB^	350.54 ± 8.23 ^bB^	0.564	0.023	0.144

Note: Different lower cases in the same line indicate significant differences between the same comparison items (*p* < 0.05), different capital letters in the same column indicate that the occurrence of the same comparison item differs significantly between day and night (*p* < 0.05), the same as below. Data are presented as mean ± SEM.

## Data Availability

The data presented in this study are available on request from the corresponding author. The data reflect the specific conditions of agricultural enterprises that are covered by the privacy policy.

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
