# Peer review of "Optimizing Feeding Strategies for Growing Rabbits: Impact of Timing and Amount on Health and Circadian Rhythms"

_animals, 2023, doi:10.3390/ani13172742_

Round 1

Reviewer 1 Report

Dear authors, 

First of all, I would like to congratulate all the authors for their extraordinary work in this section of “Animal System and Management”. You have included all the necessary information in this article. 

The manuscript was well-written and the content was informative and well-presented. I commend the authors for the comprehensive and systematic review of the topic. The manuscript will be a valuable contribution to this journal.

However, I’ve mentioned some minor corrections which need to be corrected in the comment section of the main manuscript file. Some of these include here:

1.     Please add one line at the end of the abstract, which basically explain the basic output of your study and the future recommendations related to this study work as well.

2.     Clarification of the specific research questions addressed in the study.

3.     Further recommendations mentioning which form of the specific diet you have been using for your experiment. You didn't provide any information about what type of feed you offered to the rabbits. Please provide in a tabulated format the nutritional composition and the values of the feed which have been offered to the rabbits. 

4.   Please rewrite the conclusion part of this manuscript: To highlight the basic research gap which the authors actually try to cover in this study along with their future recommendations, on the basis of their conclusion. 

5.   Please set the entire list of references according to "Animals" MDPI journal instructions. 

Best Wishes

Author Response

  1. Please add one line at the end of the abstract, which basically explain the basic output of your study and the future recommendations related to this study work as well.

Response:Following your suggestion, the following content has been added at the end of the abstract: "Our study underscores the crucial role of feeding timing in enhancing animal health. Future research should explore the underlying mechanisms and broaden the implementation of this strategy across various seasons and regions to enhance rabbit husbandry practices."

  1. Clarification of the specific research questions addressed in the study.

Response:The study aimed to explore the impact of feeding timing (whole day vs. nighttime) and feeding amount (unrestricted vs. restricted) on the growth, behavior, circadian rhythm, and overall well-being of growing rabbits in northern China during spring.

  1. Further recommendations mentioning which form of the specific diet you have been using for your experiment. You didn't provide any information about what type of feed you offered to the rabbits. Please provide in a tabulated format the nutritional composition and the values of the feed which have been offered to the rabbits. 

Response: Following your suggestion, the nutritional composition has been added. Feed during the trial was supplied by Qingdao Kangda Feed Co., Ltd., with nutritional composition detailed in Table 1. 

  1. Please rewrite the conclusion part of this manuscript: To highlight the basic research gap which the authors actually try to cover in this study along with their future recommendations, on the basis of their conclusion. 

Response:Thank you for your suggestion. The conclusion section has been revised.

“The nocturnal feeding method employed in open rabbit houses during northern China's spring aligns with the natural behavior of nocturnal rabbits. This reinforces their circadian rhythm, enhancing diurnal body temperature fluctuations and feed conversion efficiency, thereby promoting effective and healthy rabbit rearing. This study addresses the research gap in understanding the impact of feeding timing and quantity on rabbit growth, behavior, circadian rhythm, and overall well-being in northern China during spring. The authors' findings emphasize the pivotal role of feeding timing in optimizing animal health. Further exploration of the underlying mechanisms and broader implementation of this feeding strategy across diverse seasons and regions are recommended to enhance rabbit husbandry practices.”

  1. Please set the entire list of references according to "Animals" MDPI journal instructions. 

Response:Thank you for your suggestion. The entire list of references has been revised.

Modifications to the content of the article have all been highlighted using yellow markings.

Reviewer 2 Report

Dear editor and authors:

After to review the manuscript entitled “Effects of Feeding Time and Feeding Amount on the Rhythms of Behavior and Body Temperature in Growing Rabbits”. The objective of this study was determining behavior of rabbits under conditions to feeding during fattening period. The theme is interesting because there a few information about this topic, and many people are worried about the animal welfare and their repercussion in growing, performance and meat quality. Then, there are some suggestions that I would like to made.

Title. Maybe the title needs to change, because there no was informative of the content of the manuscript.

Abstract. In this section is to indicate the main findings of the study, and here there are some sentences that not important. Example: lines 16, “in northern China during spring”, in the study indicates temperature and relative humidity indoor in the facility. Lines 22 and 23, maybe this sentence is not a conclusion.

Introduction. There is a solid section, but need to specify the objective at the end of this section.

Materials and methods. First there are duplication data in Figure 1 and Table 1, need to choose one option. What is the composition of the diet? What ingredients were used to prepare diet? Lines 113 and 117 it was repeated average daily feed intake. It was indicating recompile information about incidence of mortality but later non indicates if there was a calculation of mortality percentage, but they indicate at lines 125 and 126. Need to review where is most convenient. Statistical analysis is not clear. Probably need to explain under models and specify what variable was analyzed under what model. They indicate that in some variables were used SEM, but in tables appear SD.

Results and discussion. Lines 155 to 158, sentences are not of the subtheme of rabbit performance, maybe need to explain this results in another subtheme. In Line 169 indicates the analysis of two-way ANOVA, but in the previous paragraph indicates results about the same variables. would it not be better to indicate the results at the same time?

Conclusion. Maybe need to improve the conclusion, due results and discussion of this manuscript have many others indicatives for improve this theme.

References. Maybe need to translate reference number 3.

Tables and figures. Maybe need to use SEM in the tables, figures need to be most understandable.

Author Response

  1. Title. Maybe the title needs to change, because there no was informative of the content of the manuscript.

Response: Thank you for your suggestion.  the title has been changed“Optimizing Feeding Strategies for Growing Rabbits: Impact of Timing and Amount on Health and Circadian Rhythms”

  1. In this section is to indicate the main findings of the study, and here there are some sentences that not important. Example: lines 16, “in northern China during spring”, in the study indicates temperature and relative humidity indoor in the facility. Lines 22 and 23, maybe this sentence is not a conclusion.

Response:Thank you for your suggestion. We consider it is necessary to emphasize "in northern China during spring" in the abstract, as it signifies climatic characteristics. Research indicates that variations in environmental temperature can impact animal circadian rhythms, and both winter cold stress and summer heat stress can disrupt livestock biological rhythms through cortisol or behavioral activity. Thus, we consider specifying the experimental location and season to be necessary.

Lines 22 and 23 have been revised as follows: The study aimed to investigate how feeding timing (whole day vs. nighttime) and feeding amount (unrestricted vs. restricted) affect the growth, behavior, circadian rhythm, and overall well-being of growing rabbits in northern China during the spring season.

  1. There is a solid section, but need to specify the objective at the end of this section.

Response:Following your suggestion, the following content has been added at the end of the Introduction:The study aimed to explore the impact of feeding timing (whole day vs. nighttime) and feeding amount (unrestricted vs. restricted) on the growth, behavior, circadian rhythm, and overall well-being of growing rabbits in northern China during spring.

  1. Materials and methods. First there are duplication data in Figure 1 and Table 1, need to choose one option. What is the composition of the diet? What ingredients were used to prepare diet? Lines 113 and 117 it was repeated average daily feed intake. It was indicating recompile information about incidence of mortality but later non indicates if there was a calculation of mortality percentage, but they indicate at lines 125 and 126. Need to review where is most convenient. Statistical analysis is not clear. Probably need to explain under models and specify what variable was analyzed under what model. They indicate that in some variables were used SEM, but in tables appear SD.

Response:Appreciating your suggestion, we have updated Table 1 to reflect the diet composition.

The section regarding the experimental methodology for average daily feed intake has been revised as follows:

During the experiment, daily feed intake, feed residue, incidence of mortality were recorded for all four groups, and the average daily feed intake was calculated. The rabbits in each group were weighed after a 12-hour fast at 35, 42, 49, 56, 63, 70, 77, and 84 days of age, and the average daily weight gain and feed conversion rate were calculated for each stage. At the end of the experiment,the statistical analysis encompassed the average daily feed intake, average daily gain, and feed conversion rate of rabbits throughout the entire experimental period spanning from 36 to 84 days of age.

  1. The calculation of mortality percentage has been added to the Results section, as indicated in line 202.

Response:Throughout the experiment spanning from 36 to 84 days of age, the mortality rates of 108 rabbits in each group were as follows: NRF, WRF, NUF, and WUF groups exhibited rates of 4.63%, 4.63%, 0.96%, and 2.78% respectively.

  1. Results and discussion. Lines 155 to 158, sentences are not of the subtheme of rabbit performance, maybe need to explain this results in another subtheme. In Line 169 indicates the analysis of two-way ANOVA, but in the previous paragraph indicates results about the same variables. would it not be better to indicate the results at the same time?

Response:Thank you for your suggestion. We have integrated the results of the two-way ANOVA analysis with the results indicated in the previous paragraph.

  1. Maybe need to translate reference number 3.

Response:Thank you for your suggestion. The references have been revised.

  1. Tables and figures. Maybe need to use SEM in the tables, figures need to be most understandable.

Response:Following your suggestion, relevant information has been incorporated into the captions of the charts and tables.

Modifications to the content of the article have all been indicated using yellow markings.

Reviewer 3 Report

The main shortcoming of the manuscript is the number of rabbits observed – only three per group and the frequency of rectal temperature examination. I recommend the authors in such type of studies to observe the behavior of at least 6-10 rabbits and for body temperature to use infrared thermography instead of rectal temperature, which is much more efficient and more humane than rectal temperature.

Author Response

Thank you for your suggestion. As you mentioned, there is indeed a shortage of video data in the current experiment. We will address this issue in subsequent trials. In addition, regarding the real-time monitoring of body temperature, the infrared detection method is not accurate and is influenced by factors such as rabbit fur, test location, and detection distance, leading to significant variations in values and an inability to obtain accurate temperature parameters.

Round 2

Reviewer 3 Report

Manuscript submitted to me for review, „Effects of Feeding Time and Feeding Amount on the Rhythms of Behavior and Body Temperature in Growing Rabbits“, examines the effects of feeding time and amount of feeding on behavior and body temperature in rabbits. The idea for the article is relevant because it tries to solve a problem with rabbit breeding in industrial conditions, namely its physiological, respectively health condition. The main contribution of the study is that it examines several feeding regimes and amount of feed on the behavior and health of rabbits during the variable temperature conditions of spring.